# AirID, a novel proximity biotinylation enzyme, for analysis of protein–protein interactions

**Kohki Kido[1], Satoshi Yamanaka[1], Shogo Nakano[2], Kou Motani[3], Souta Shinohara[1], Akira Nozawa[1], Hidetaka Kosako[3], Sohei Ito[2], Tatsuya Sawasaki[1]\***

[1]Division of Cell-Free Life Science, Proteo-Science Center, Matsuyama, Japan; [2]Graduate School of Integrated Pharmaceutical and Nutritional Sciences, University of Shizuoka, Shizuoka, Japan; [3]Division of Cell Signaling, Fujii Memorial Institute of Medical Sciences, Tokushima University, Tokushima, Japan

**Abstract** Proximity biotinylation based on *Escherichia coli* BirA enzymes such as BioID (BirA\*) and TurboID is a key technology for identifying proteins that interact with a target protein in a cell or organism. However, there have been some improvements in the enzymes that are used for that purpose. Here, we demonstrate a novel BirA enzyme, AirID (ancestral BirA for proximity-dependent biotin identification), which was designed de novo using an ancestral enzyme reconstruction algorithm and metagenome data. AirID-fusion proteins such as AirID-p53 or AirID-IκBα indicated biotinylation of MDM2 or RelA, respectively, in vitro and in cells, respectively. AirID-CRBN showed the pomalidomide-dependent biotinylation of IKZF1 and SALL4 in vitro. AirID-CRBN biotinylated the endogenous CUL4 and RBX1 in the CRL4$^{CRBN}$ complex based on the streptavidin pull-down assay. LC-MS/MS analysis of cells that were stably expressing AirID-IκBα showed top-level biotinylation of RelA proteins. These results indicate that AirID is a novel enzyme for analyzing protein–protein interactions.

**\*For correspondence:** sawasaki@ehime-u.ac.jp

**Competing interests:** The authors declare that no competing interests exist.

## Introduction

Many cellular proteins function under the control of biological regulatory systems. Protein–protein interactions (PPIs) comprise part of the biological regulation system for proteins. Besides PPIs, biological protein function is post-translationally promoted by multiple modifications such as complex formation, phosphorylation, and ubiquitination. Therefore, it is very important to understand how proteins interact with target proteins. The identification of partner proteins has been carried out using several technologies such as the yeast two-hybrid system (*Zhao et al., 2017*; *Li et al., 2016*), mass spectrometry analysis after immunoprecipitation (*Ohshiro et al., 2010*; *Han et al., 2015*), and cell-free-based protein arrays that we have previously described (*Nemoto et al., 2017*; *Takahashi et al., 2016*). These methods provide many critical findings. As intracellular proteins are regulated by quite complicated systems, such as signaling transduction cascades, the use of multiple technologies can strongly promote our understanding of cellular protein regulation.

Proximity-labelling technology has been widely used to identify partner proteins (*Chang et al., 2017*). As proximity labelling is thought to detect proteins that are very close together, it is expected to obtain more precise information about interacting proteins (*Kim et al., 2014*). Currently, EMARS (enzyme-mediated activation of radical sources) (*Honke and Kotani, 2012*; *Kotani et al., 2008*), APEX (engineered ascorbate peroxidase) (*Martell et al., 2012*; *James et al., 2019*), and BioID (the proximity-dependent biotin identification) *Choi-Rhee et al., 2004*; *Roux et al., 2012* have been developed as proximity-labelling technologies. EMARS and APEX

**eLife digest** Proteins in a cell need to interact with each other to perform the many tasks required for organisms to thrive. A technique called proximity biotinylation helps scientists to pinpoint the identity of the proteins that partner together. It relies on attaching an enzyme (either BioID or TurboID) to a protein of interest; when a partner protein comes in close contact with this construct, the enzyme can attach a chemical tag called biotin to it. The tagged proteins can then be identified, revealing which molecules interact with the protein of interest.

Although BioID and TurboID are useful tools, they have some limitations. Experiments using BioID take more than 16 hours to complete and require high levels of biotin to be added to the cells. TurboID is more active than BioID and is able to label proteins within ten minutes. However, under certain conditions, it is also more likely to be toxic for the cell, or to make mistakes and tag proteins that do not interact with the protein of interest.

To address these issues, Kido et al. developed AirID, a new enzyme for proximity biotinylation. Experiments were then conducted to test how well AirID would perform, using proteins of interest whose partners were already known. These confirm that AirID was able to label partner proteins in human cells; compared with TurboID, it was also less likely to mistakenly tag non-partners or to kill the cells, even over long periods.

The results by Kido et al. demonstrate that AirID is suitable for proximity biotinylation experiments in cells. Unlike BioID and TurboID, the enzyme may also have the potential to be used for long-lasting experiments in living organisms, since it is less toxic for cells over time.

methods produce $H_2O_2$ as a label (*Kotani et al., 2008*; *Martell et al., 2012*). They can rapidly label protein, but $H_2O_2$ is highly cytotoxic (*Halliwell et al., 2000*).

At present, proximity biotinylation is based on the *Escherichia coli* enzyme, BirA. BioID (proximity-dependent biotin identification) was first reported in 2004, and its main improvement was the single BirA mutation at R118G (BirA*) (*Choi-Rhee et al., 2004*). BioID generally has promiscuous activity and releases highly reactive and short-lived biotinoyl-5′-AMP. Released biotinoyl-5′-AMP modifies proximal proteins (within a distance of 10 nm) (*Kim et al., 2014*). BioID can be used by expressing the BioID-fusion protein and adding biotin. In cells expressing BioID-fusion bait protein, proteins with which the bait protein interacts are biotinylated and can be comprehensively analyzed using precipitation with streptavidin followed by mass spectrometry (*Roux et al., 2012*). BioID can easily analyze the protein interactome in mild conditions. However, BioID takes a long time (>16 hr) and requires a high biotin concentration to biotinylate interacting proteins. Therefore, it cannot easily detect short-term interactions and is difficult to use in vivo. Second, BioID was improved using R118S and 13 mutations via yeast-surface display; this yielded TurboID (*Branon et al., 2018*). TurboID has extremely high activity and can biotinylate proteins in only ten minutes. However, TurboID caused non-specific biotinylation and cell toxicity when labeling times were increased and biotin concentrations were high (*Branon et al., 2018*). In addition, a small BioID enzyme from *Aquifex aeolicus* was reported as BioID2 (*Kim et al., 2016*). BioID, TurboID, and BioID2 are excellent enzymes, and they offer some improvements for the proximity biotinylation of cellular target proteins. Further improvement of BirA enzymes is an important goal that would enhance the convenience of proximity biotinylation in cells.

Evolutionary protein engineering using metagenome data have recently been used to improve enzymes (*Nakano and Asano, 2015*; *Nakano et al., 2018*; *Nakano et al., 2019*). Here, we newly designed five ancestral BirA enzymes using an ancestral enzyme reconstruction algorithm and a large genome dataset. The combination of ancestral reconstruction and site-directed mutagenesis has provided a newly useful BirA enzyme, AirID (ancestral BirA for proximity-dependent biotin identification), which functions in proximity biotinylation in vitro and in cells. Although the sequence similarity between BioID and AirID is 82%, AirID showed high biotinylation activity against interacting proteins. Our results indicate that AirID is a useful enzyme for analyzing protein–protein interactions in vitro and in cells.

## Results

### Reconstruction of ancestral BirA enzyme using metagenome data

BioID and TurboID were designed on the basis of the biotin ligase BirA from *E. coli*. Using a different approach, we attempted to reconstruct the ancestral BirA sequence. Five ancestral sequences were obtained using the following process. A comprehensive and curated sequence library was prepared querying the Blastp web server and using a custom Python script (*Source code 1*), which exhibited more than 30% sequence identity with *E. coli* BirA (EU08004.1). Next, further curation approaches were applied to the library as shown in previous studies (*Nakano et al., 2018*; *Nakano et al., 2019*). The procedure consists of the following steps: 1) preparation of sequence pairs consisting of one of the submitted BirA sequences and one sequence (total 1275 genes) in the library, 2) sequence alignment of the all pairs, and 3) selection of sequences bearing 'key residues'

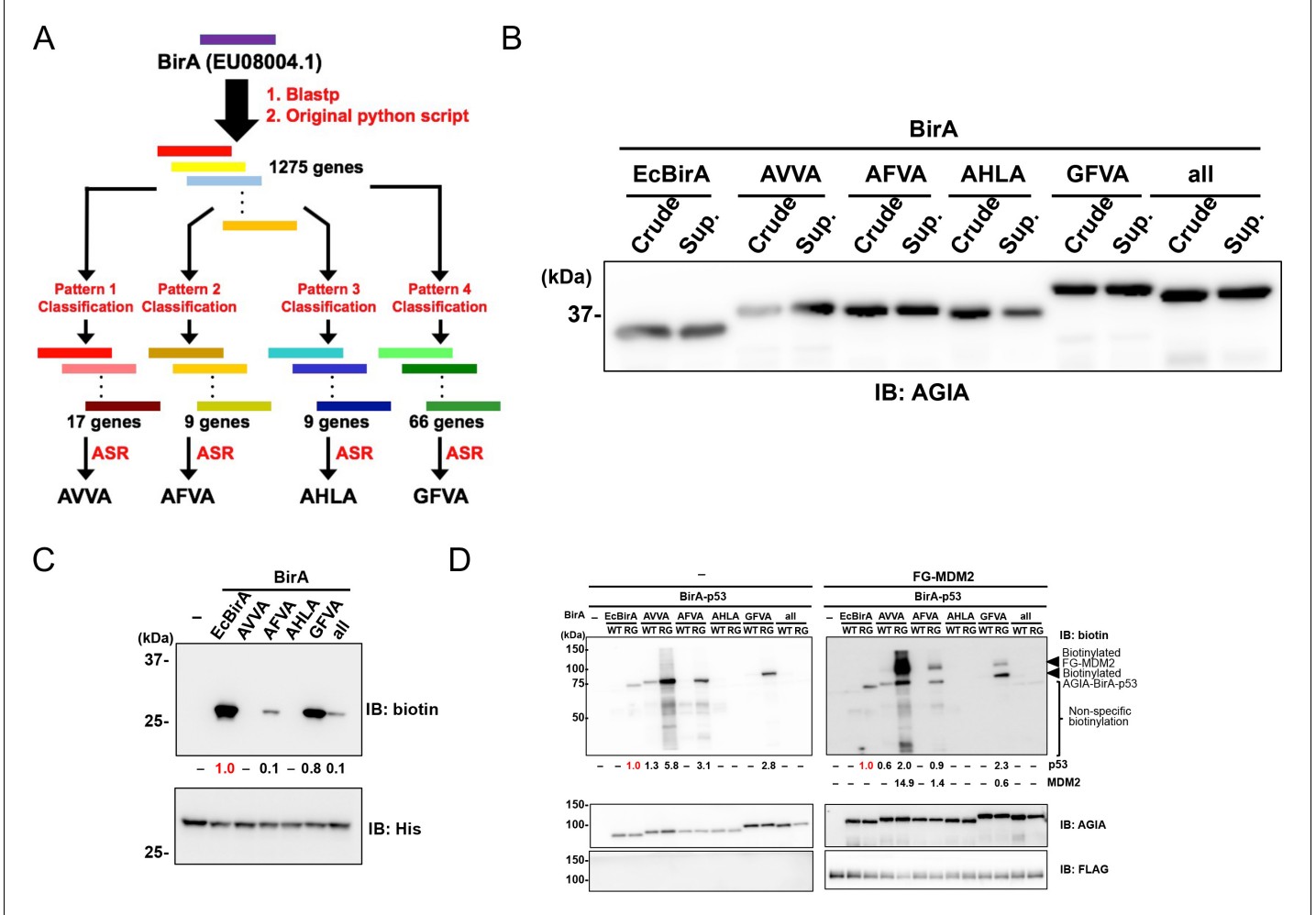

**Figure 1.** Characterization of novel BirA enzymes designed using metagenome data. (**A**) A homolog library of BirA from *E. coli* (EcBirA) was generated using blastp and curated using an original python script. The curated library was multiple aligned using INTMSAlign and sequences were classified into four groups. Each group was phylogenetically analyzed, and ancestral sequences were designed. (**B**) AGIA-tagged AncBirAs were synthesized using the wheat cell-free system. Their expressions were confirmed using anti-AGIA antibody immunoblotting. (**C**) EcBirA or each AncBirA was added to the reaction mixture when His-bls-FLAG-GST was synthesized. Biotinylation of bls by each BirA was examined using anti-biotin antibody immunoblotting. As a control, the expression of each BirA was detected using His antibody. The band intensity of biotinylated His-bls-FLAG-GST was quantified with image J software, with the index intensity (value 1.0) shown in in red characters. (**D**) The WT or RG mutant of each BirA was fused to p53 (BirA-p53). They analyzed biotinylation activity with or without FLAG-GST-MDM2. As a control, the expression of each BirA-p53 and MDM2 was detected using an anti-AGIA antibody and an anti-FLAG antibody, respectively. The band intensities of biotinylated p53 and MDM2 were quantified with image J software. The index intensity (value 1.0) is shown in red characters.

(*Figure 1A*). In detail, we prepared the following four combinations of the key 26th, 124th, 171th, and 297th residues to classify the library: Ala, Val, Val, Ala (pattern 1, AVVA); Ala, Phe, Val, Ala (pattern 2, AFVA); Ala, His, Leu, Ala (pattern 3, AHLA); and Gly, Phe, Val, Ala (pattern 4, GFVA) (*Figure 1A*). After the selection, we classified the library as follows: the library could be divided into 17, 9, 9, or 66 genes depending on whether the key residues consisted of pattern 1, 2, 3, or 4, respectively (*Figure 1*). Utilizing each of the classified genes, we designed four artificial sequences using the ancestral sequence reconstruction (ASR) method (*Supplementary file 1*). The designed sequences were named on the basis of the patterns; the sequences were classified using the patterns 1 to 4, which we refer to as AVVA, AFVA, AHLA, and GFVA, respectively (*Figure 1A*). Furthermore, we added an 'all' BirA enzyme from the common ancestor of AVVA, AFVA, and GFVA. BirA enzymes of AVVA, AFVA, AHLA, and GFVA shared similarity with those in the *Shewanella* genus, the *Frischella* and *Glliamella* genera, the *Thiobacillus* and *Betaproteobacteria* genera, and multiple genera, respectively. When the AVVA, AFVA, AHLA, GFVA, and 'all' amino-acid sequences were compared to the sequence *E. coli* BirA they showed 45%, 58%, 42%, 82%, and 73% similarity, respectively, and the region including the active site (107–134 amino acids) was the same sequence throughout.

## Enzymatic characterizations of newly designed ancestral BirA enzymes

On the basis of the amino-acid sequences discussed above, five DNA templates for AVVA, AFVA, AHLA, GFVA, and 'all' were prepared using artificial DNA synthesis. To convert the designed proteins into DNA sequences, we used the codon usage profile from the plant *Arabidopsis*, the average genic AT content of which is nearly 50% (*Arabidopsis Genome Initiative, 2000*). All ancestral BirA genes were fused an N-terminal AGIA tag, because this is a highly sensitive tag based on a rabbit monoclonal antibody (*Yano et al., 2016*). We synthesized these ancestral BirA proteins (AncBirAs) using a wheat cell-free protein production system to investigate their enzymatic potentials (*Sawasaki et al., 2002*). Biotin ligase activity was subsequently checked as all ancestral BirA proteins were obtained as a soluble form (*Figure 1B*). A His-bls-FLAG-GST protein with a N-terminal biotinylation site (bls) of GLNDIFEAQKIEWHE for *E. coli* BirA (EcBirA) was used as a substrate. Three ancestral BirA proteins—AFVA, GFVA, and 'all'—showed activity against the bls sequence (*Figure 1C*). GFVA had the greatest activity, similar to that of EcBirA, whereas AVVA and AHLA did not have activity.

In all of the designed ancestral BirA proteins, an arginine residue corresponding to R118 of EcBirA was conserved in an active site for biotinylation (*Supplementary file 1*). Because EcBirA gained proximity biotinylation activity as a result of the R118G mutation known as BioID (*Choi-Rhee et al., 2004*), each R residue in the five genes was substituted with glycine (RG mutants). To compare the proximity biotinylation activity among these genotypes, wild-type or RG mutant BirA gene was N-terminally fused to the p53 gene. The resulting BirA-p53 proteins were synthesized using the cell-free system before mixing with FLAG-GST(FG)-MDM2, because an interaction between p53 and MDM2 has been widely observed (*Momand et al., 1992*; *Michael and Oren, 2003*). Immunoblotting revealed BirA-p53 biotinylation in EcBirA-RG (BioID), AVVA-WT, AVVA-RG, AFVA-RG, and GFVA-RG (*Figure 1D*). FG-MDM2 proximity biotinylation was detected under these conditions in three ancestral BirA-RG mutants—AVVA-RG, AFVA-RG, and GFVA-RG—indicating that they are candidate enzymes for proximity biotinylation.

AVVA-RG showed both the highest activity of proximity biotinylation and extra biotinylations in the lower size region ('non-specific biotinylation' in *Figure 1D*) when compared to the three ancestral BirA-RG mutants. GFVA-RG indicated the highest biotinylation activity for a specific peptide (*Figure 1C*) and the lowest extra proximity biotinylation. According to these results, we focused on two enzymes, AVVA and GFVA, for further analysis.

## Proximity biotinylation ability of the ancestral BirA-RS mutants under different conditions

TurboID was recently reported as an improved BioID enzyme (*Branon et al., 2018*). As TurboID has an R118S mutation (RS mutant) that increases the activity of proximity biotinylation, we made RS mutants of the two ancestral BirA enzymes and compared their proximity biotinylation activities in vitro and in cells. An interaction between N-terminal AGIA-BirA-fusion p53 (AGIA-BirA-p53) and FG-MDM2 was used to validate proximity biotinylation ability in vitro. Incubation time, biotinylation

temperature, and biotin concentration were investigated as conditions for proximity biotinylation. Consequently, TurboID, AVVA-RG, AVVA-RS, and GFVA-RS showed higher proximity biotinylation activity after 3 hr than did BioID with a 16-hr incubation (*Figure 2A*). The GFVA enzyme with a RS mutation dramatically increased the activity of proximity biotinylation to RelA (GFVA-RS in *Figure 2A* and *Figure 2—figure supplement 1*), but proximity biotinylation was almost the same in AVVA-RG and -RS. AVVA-RG, AVVA-RS, and GFVA-RS showed high activity of proximity biotinylation at temperatures above 16 ℃ (*Figure 2—figure supplement 1A*). AVVA-RG, AVVA-RS, and GFVA-RS showed high proximity biotinylation activity at biotin concentrations greater than 0.5 µM (*Figure 2—figure supplement 1B*). On the basis of these results, three BirA enzymes—AVVA-RG, AVVA-RS, and GFVA-RS—were used for further analysis.

We used IκBα and RelA to validate the proximity biotinylation ability of these three enzymes in other protein–protein interactions because the IκBα–RelA interaction has been widely observed (*Beg et al., 1992*; *Baeuerle and Baltimore, 1988*). As in the analysis of the p53–MDM2 interaction, N-terminal AGIA-BirA-fusion IκBα (AGIA-BirA-IκBα) and FLAG-GST-RelA (FG-RelA) sequences were constructed. FG-MDM2 was used as a negative control. To compare the abilities of the different enzymes directly, the reactions of all enzymes were carried out under the same conditions. After co-incubating AGIA-BirA-IκBα and FG-RelA, AVVA-RS or GFVA-RS, high RelA biotinylation was indicated (*Figure 2B*). FG-MDM2 biotinylation by AGIA-BirA-IκBα was not observed.

## Proximity biotinylation of the ancestral BirA-RS mutants in cells

Next, the proximity biotinylation ability of these three enzymes was validated in cells. MDM2 dramatically degrades p53 protein in cells (*Michael and Oren, 2003*), so a CS mutant (MDM2(CS)) lacking E3 ubiquitin ligase activity was used for this assay. In addition, GFP (green fluorescent protein) was terminally fused to MDM2 (GFP-MDM2(CS)) because the mobility size of this fusion protein on SDS-PAGE is very similar to that of BirA-fusion p53 and MDM2. AGIA-BirA-p53 fusions were transiently expressed in HEK293T cells with or without GFP-MDM2(CS), and they were compared with or without biotin supplement. GFVA-RS showed higher biotinylation of FG-MDM2 than did other enzymes under conditions without biotin supplementation (left panel in *Figure 2—figure supplement 2*). Furthermore, GFVA-RS also indicated biotinylation of FG-MDM2 under biotin supplementation conditions (right panel). Taken together, these results indicated that an ancestral BirA with GFVA-RS is a good enzyme for analyzing protein–protein interactions both in vitro and in cells. Thus, we selected GFVA-RS, and we called this AirID (ancestral BirA for proximity-dependent biotin identification, an homage to BioID and TurboID).

## Biochemical characterization of the AirID (GFVA-RS) enzyme

Before utilizing AirID for various applications, we assessed AirID for the two activities self-biotinylation and 5′-biotinyl-AMP production, because these activities indicate proximity biotinylation. It is known that the p53 protein makes a homo multimer (*Friedman et al., 1993*; *Delphin et al., 1994*). Each enzyme alone or the p53-fusion form was used to investigate the two activities in BioID, TurboID, and AirID. BioID and TurboID showed self biotinylation, and TurboID had the highest activity (*Figure 2—figure supplement 3*). AirID did not have the activity, indicating that AirID does not self-biotinylate. As TurboID was selected as an enzyme by screening the yeast-surface display that showed the highest self-biotinylation activity (*Branon et al., 2018*), the highest activity from TurboID is reasonable. The lack of self-biotinylation activity in AirID may be caused by a property of AirID as the enzyme or by a lack of accessible lysine residues on AirID.

We next investigated the ability of the AirID enzyme to produce biotinoyl-5′-AMP. His-tagged TurboID, GFVA-RG, and AirID proteins were produced in an *E. coli* system and purified using nickel sepharose beads. Highly purified enzymes were obtained (*Figure 2—figure supplement 4A*) and biotinylation of TurboID was found, indicating that TurboID biotinylated itself in *E. coli* cells. As shown in *Figure 2—figure supplement 3*, self-biotinylation of AirID was not observed. Furthermore, to investigate the biotinylation ability of AirID at the biotin ligation site (bls) of *E. coli* BirA, purified His-tag and bls fusion FLAG-GST protein (His-bls-FLAG-GST) was used as a substrate. AirID and GFVA-RG biotinylated His-bls-FLAG-GST (*Figure 2—figure supplement 4B*), but TurboID did not do so. Radio-isotope-labelled ATP [$^{32}$P-α-ATP] was used according to a previous report (*Henke and*

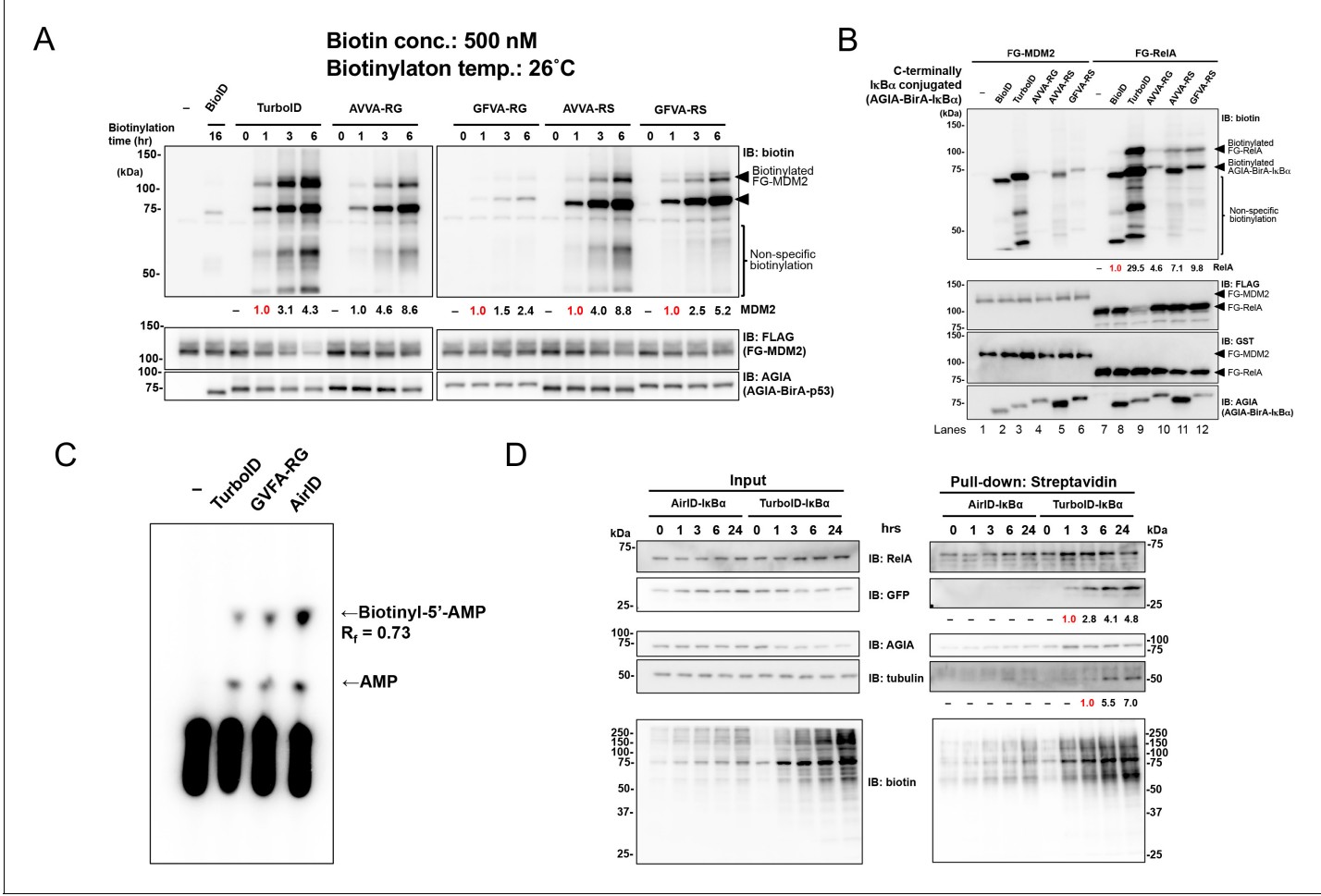

**Figure 2.** Validation of PPI dependency of novel designed BirA enzymes. (**A**) RS mutants of AVVA and GFVA were cloned, and biotinylations of FLAG-GST-MDM2 (FG-MDM2) by BirA-p53 including RS mutants were analyzed. The reaction was performed at 500 nM of biotin at 26 °C for the described time. As a control, the expression levels of both BirA-p53 and MDM2 were detected using anti-AGIA antibody and anti-FLAG antibody, respectively. The band intensity of biotinylated MDM2 was quantified with image J software. The index intensity (value 1.0) is shown in red characters. (**B**) FG-RelA biotinylation by BirA-IκBα was examined. FG-MDM2 was used as the negative control. Biotinylations were performed at 500 nM of biotin at 26 °C for 1 hr (TurboID), 3 hr (AVVA-RG, AVVA-RS, and GFVA-RS), or 16 hr (BioID). As a control, the expression levels of BirA-p53 and MDM2 were detected using anti-AGIA antibody, anti-FLAG antibody and anti-GST antibody. The band intensity of biotinylated RelA was quantified with image J software. The index intensity (value 1.0) is shown in red characters. (**C**) GFVA-RG and GFVA-RS expressed using *E. coli* were purified using Ni beads and mixed with His-bls-FLAG-GST, which was synthesized using a wheat cell-free system and purified using glutathione beads. The mixtures were incubated a solution including [α-$^{32}$P]ATP and biotin for 30 min at 37 °C. The resultant biotinyl-5'-AMP, AMP, or unreacted ATP was separated using cellulose thin-layer chromatography. (**D**) GFP and either AirID-IκBα or TurboID-IκBα were transfected in HEK293T, and biotin was added to 5 µM of this mixture for the described time period. After transfecting for 24 hr, cells were lysed by RIPA buffer including protease inhibitors, and biotinylated proteins were pulled down with streptavidin beads. As a control, the expression levels of enzyme-fused protein and target proteins were detected using each protein-specific antibody (left panel). The band intensity of pulled-down GFP and tubulin was quantified with image J software. The index intensity (value 1.0) is shown in red characters.

The online version of this article includes the following figure supplement(s) for figure 2:

**Figure supplement 1.** Condition for in vitro biotinylation.
**Figure supplement 2.** Comparison of MDM2 biotinylation by BirA-p53 enzymes.
**Figure supplement 3.** Self-biotinylation activity of novel designed BirA enzymes.
**Figure supplement 4.** Comparison of enzymatic activity among GFVA-RG, AirID (GFVA-RS), and TurboID.
**Figure supplement 5.** Optimization of AirID-dependent biotinylation in cells.

Cronan, 2014) to detect biotinoyl-5′-AMP production by the enzymes. The ATP concentration reported by this assay was very low (final 1 µM) because of the use of labelled ATP. AirID and GFVA-RG produced biotinoyl-5′-AMP, and this activity was decreased by supplementing with His-bls-FLAG-GST (Figure 2C). AMP was increased at the same time, indicating that biotinoyl-5′-AMP is

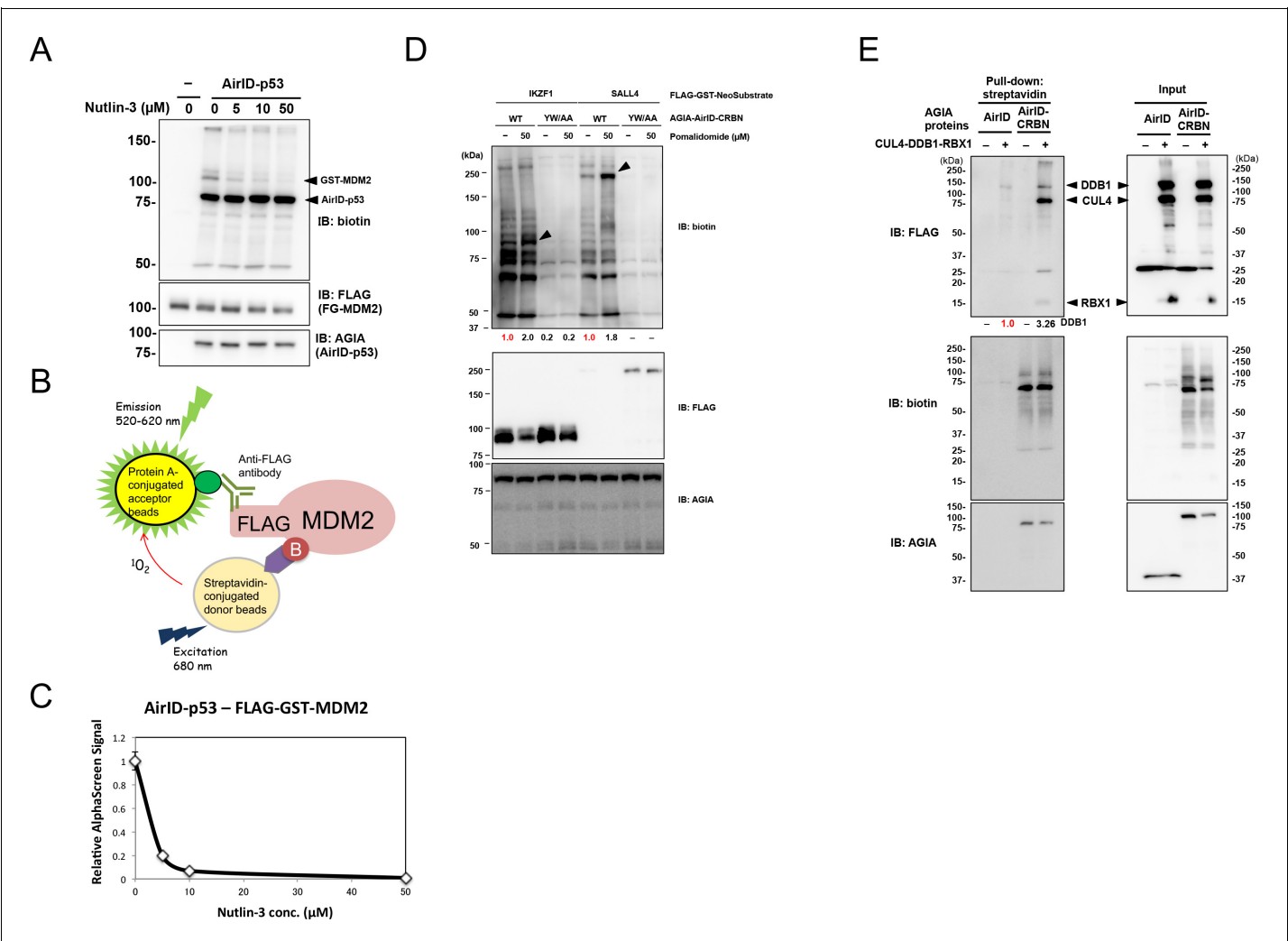

**Figure 3.** Biochemical applications of AirID-dependent biotinylation on PPI. (**A**) Biotinylations of FG-MDM2 by AirID-p53 were carried out with or without Nutlin-3, which inhibits the interaction between p53 and MDM2, at 500 nM of biotin at 26 ℃ for 3 hr. Biotinylated MDM2 was detected using immunoblotting. As a control, expression levels of BirA-p53 and MDM2 were detected using anti-AGIA antibody and anti-FLAG antibody, respectively. (**B**) MDM2 biotinylation was detected using AlphaScreen with the reaction mixtures described for panel (**A**). Biotinylated FG-MDM2 interacts with both streptavidin donor beads and protein A acceptor beads to which the anti-FLAG antibody binds. The AlphaScreen results are shown in panel (**C**). (**D**) Pomalidomide-dependent biotinylations of FG-IKZF1 and FG-SALL4 by AirID-CRBN were analyzed. FG-IKZF1 or FG-SALL4 was biotinylated with or without pomalidomide at 500 nM of biotin at 26 ℃ for 3 hr. As the negative control, YW/AA mutant of AirID-CRBN, which does not bind to pomalidomide, was used. As a control, expression of AirID-CRBN and IKZF1 or SALL4 was detected using anti-AGIA antibody and anti-FLAG antibody, respectively. The band intensity of biotinylated IKZF1 or SALL4 was quantified with image J software. The index intensity (value 1.0) is shown in red characters. (**E**) CRL4$^{CRBN}$ complex proteins were biotinylated using AirID or AirID-CRBN. Biotinylated proteins were pulled down with streptavidin beads. As a control, the expression levels of AirID-CRBN and the complex component proteins was detected using anti-AGIA antibody and anti-FLAG antibody, respectively (right panel). The band intensity of biotinylated DDB1 was quantified with image J software. The index intensity (value 1.0) is shown in red characters.

The online version of this article includes the following source data and figure supplement(s) for figure 3:

**Source data 1.** AlphaScreen data used to generate *Figure 3C*.

**Figure supplement 1.** CRBN-dependent biotinylation of IKZF1 and SALL4 with pomalidomide.

**Figure supplement 2.** FT-dependent biotinylation of FD in the co-translational condition, based on the cell-free system.

produced by these enzymes and that AMP is released after biotinylation of His-bls-FLAG-GST. Bioti-noyl-5′-AMP formation was shown as AirID > GFVA-RG > TurboID by comparing the enzymes under these conditions (*Figure 2—figure supplement 4C*), suggesting that AirID has higher biotinoyl-5′-AMP formation than TurboID under low ATP conditions.

## Proximity biotinylation conditions of AirID in cells

We compared the optimal conditions for proximity biotinylation in cells between BioID, TurboID, and AirID. As a model of proximity biotinylation, AGIA-BirA-fusion p53 and FG-MDM2(CS) were co-expressed in cells. Biotin concentration and biotinylation time were investigated as the variable conditions for proximity biotinylation. Consequently, AirID and TurboID biotinylated MDM2 at biotin concentrations higher than 0.5 µM within 3 hr and 1 hr in cells, respectively (*Figure 2—figure supplement 5A and B*). Although TurboID-p53 dramatically increases biotinylation of high molecular weight products with long incubations of >6 hr with >5 µM biotin, AirID-p53 showed similar results from 3 to 24 hr and with 0.5–50 µM biotin supplementation in culture medium. This indicates that the AirID-fusion protein could function in a wide variety of conditions.

Furthermore, PPI dependency was examined between AirID-IκBα and TurboID-IκBα. GFP and either AGIA-AirID-IκBα or AGIA-TurboID-IκBα were coexpressed in HEK293T cells. Next, biotinyla-tion after 1, 3, 6, and 24 hr incubation with 5 µM of biotin supplementation was analyzed using a streptavidin-pull down assay. Each protein was detected using each specific antibody. As shown in the input sample (left panel in *Figure 2D*), expression of both fusion enzymes was at nearly the same level (IB: AGIA). TurboID showed much higher biotinylation in whole lysates than did AirID. In the pull-down assay, both enzymes biotinylated endogenous RelA at all points. After 1 hr of incubation, biotinylation of co-expressed GFP was found in TurboID-IκBα, and continuous tubulin biotinylation was also carried out after 3 hr (right panel in *Figure 2D*). They found no AirID-IκBα biotinylation, although AirID was incubated for 24 hr with biotin. These results indicated that AirID has high PPI dependency.

## Biochemical applications of AirID-dependent biotinylation in protein–protein interaction

We used AirID for various in vitro applications. It has been widely known that p53–MDM2 interaction is inhibited by nutlin-3 (*Vassilev et al., 2004*). Nutlin-3 was used to investigate whether AirID can be used to validate an inhibitor of PPI. Immunoblotting revealed that nutlin-3 inhibited FG-MDM2 bioti-nylation by AGIA-AirID-p53 (*Figure 3A*). As we used AlphaScreen technology for drug screening of PPI in previous reports (*Uematsu et al., 2018*; *Nemoto et al., 2018*; *Nomura et al., 2019*; *Yamanaka et al., 2020*), we used it to detect the biotinylation of drug-dependent PPI inhibition (*Figure 3B*). FG-MDM2 biotinylation by AGIA-AirID-p53 interaction was also detected (0 µM in *Figure 3C*) using AlphaScreen technology, and the signal was decreased by supplementing with nutlin-3 (>10 µM). These results indicate that AirID can detect PPI inhibition by the drug.

Thalidomide and its derivatives such as pomalidomide bind to cereblon (CRBN) (*Ito et al., 2010*; *Lopez-Girona et al., 2012*) and degrade target proteins such as IKZF1 (*Lu et al., 2014*; *Krönke et al., 2014*) and SALL4 (*Matyskiela et al., 2018*) in cells. These small chemical compounds are known as molecular glue (*Fischer et al., 2016*). As the CRBN-YW/AA mutant loses thalidomide binding ability (*Ito et al., 2010*), it could not interact with IKZF1 and SALL4 proteins. To investigate whether AirID detects a drug-dependent PPI in vitro, CRBN–IKZF1 and CRBN–SALL4 interactions were analyzed with or without pomalidomide. Biotinylations of FG-IKZF1 and FG-SALL4 by AGIA-AirID-CRBN (WT) were increased by supplementing with pomalidomide (*Figure 3D*). However, bioti-nylations were not found in CRBN-YW/AA. These results indicate that AirID can detect a drug-dependent PPI in vitro. Furthermore, the biotinylations of neosubstrates by AirID-CRBN with pomali-domide were investigated in cells. Myc-tag fusion IKZF1 (myc-IKZF1) and myc-SALL4 were transiently co-expressed with AGIA-AirID-CRBN (WT) or AGIA-AirID-CRBN-YW/AA in CRBN-knockout HEK293T cells. By supplementing with 5 µM biotin, biotinylations of both myc-IKZF1 and myc-SALL4 were detected by expressing AGIA-AirID-CRBN (WT) (*Figure 3—figure supplement 1*). They were not observed in co-expression with AGIA-AirID-CRBN-YW/AA. These results indicate that AirID can detect a drug-dependent PPI in cells.

CRBN is involved in the Cullin-4 complex consisting of DDB1, RBX1, and CUL4 (*Fischer et al., 2014*). To investigate whether AirID detects proteins in a multiple complex, AirID-CRBN was mixed in with the complex members. Biotinylations of DDB1, CUL4, and RBX1 by AGIA-AirID-CRBN were observed (*Figure 3E*), but AGIA-AirID was not biotinylated. This indicated that AirID can detect PPI in a multiple protein complex.

The Flowering locus T (FT) protein, known as the flowering hormone florigen in plants, induces the differentiation of flowering with Flowering locus D (FD) protein, which has a bZip DNA-binding domain (*Abe et al., 2005*). FT–FD interaction in the floral meristem has been thought to be an important event for flowering development (*Jaeger et al., 2006*). To investigate whether the FT–FD interaction detects biotinylation of AirID, FT and FD genes in *Arabidopsis* were selected to co-synthesize FT-AirID and AGIA-FD proteins using by the wheat cell-free system with 500 nM biotin. As a negative control, *E. coli* dihydrofolate reductase (DHFR) was synthesized with FT-AirID. Under these conditions, FT-AirID biotinylated AGIA-FD , but AGIA-DHFR biotinylation was not observed (*Figure 3—figure supplement 2*). This co-translational condition reaction was incubated for 16 hr at 16 °C with biotin, indicating that AirID-dependent biotinylation functions in co-translational conditions based on the cell-free system. Taken together, these results indicate that the AirID enzyme is useful for biochemical analysis of PPI.

## Cellular localization of AirID and AirID-p53

We next analyzed the cellular localization of AirID and cellular biotinylation by AirID. The p53 protein is known to localize mainly to the nucleus (*Shaulsky et al., 1990*; *Rotter et al., 1983*). AGIA-AirID alone or AGIA-AirID-p53 was transiently expressed in HEK293T cells. Fluorescent streptavidin was found in whole cells by supplementing AirID expression cells with biotin (50 µM in *Figure 4A*). In AirID-p53 expression cells, the fluorescence was mainly observed in the nucleus, and it was at the same level for cells exposed to either 5 µM or 50 µM biotin concentration (*Figure 4A*). Cellular fractions from cytosol and nuclei were isolated to confirm the cellular localization by immunostaining. These fractionations indicated that AirID and AirID-p53 were mainly found in the cytosol or nucleus, respectively (*Figure 4B*). This suggested that AirID-fusion protein localization is dependent on fusion protein features.

## Functions of proteins biotinylated by AirID in cells

We investigated whether proteins that were biotinylated by AirID have native function. Because MDM2 has been known to induce p53 degradation via the ubiquitin-proteasome system in cells (*Michael and Oren, 2003*) and AirID-p53 provided both self and MDM2 biotinylation (*Figure 2A*), AGIA-AirID-p53 and GFP-MDM2 were transiently co-expressed in cells. Treatment with proteasome inhibitor MG132 inhibited AirID-p53 degradation, but degradation was extremely decreased without the treatment (*Figure 4C*). The inactive MDM2 form, FG-MDM2(CS), also did not promote AirID-p53 (*Figure 2A*), indicating that AirID-p53 degradation is carried out by GFP-MDM2. In addition, MG132 treatment biotinylated AirID-p53 under biotin supplementation conditions. These results indicated that biotinylated MDM2 works as a E3 ligase for biotinylated AirID-p53.

RelA was selected to investigate the transactivation activity of the biotinylated transcription factor because RelA has transactivation activity for the NF-κB promoter (*Ganchi et al., 1992*). Two types of expression plasmids, AGIA-AirID-RelA and AGIA-RelA, were constructed. Each plasmid was transiently transfected in HEK293 cells with a NF-κB promoter-luciferase plasmid. Biotin supplementation induced biotinylation of AGIA-AirID-RelA, but AGIA-RelA biotinylation was not found (*Figure 4D*). The luciferase assay revealed that the transactivation activity of AirID-RelA was nearly the same for AGIA-RelA, AGIA-AirID-RelA, and biotinylated AGIA-AirID-RelA, indicating that biotinylated RelA functions as a normal transcription factor.

## AirID effects on cell viability

TurboID almost completely inhibits HEK293T cell growth under 50 µM biotin supplementation conditions (*Branon et al., 2018*). HEK293T cells that stably expressed AirID or AirID-IκBα were constructed using a lentivirus system to investigate whether AirID affects HEK293T cell viability. In the stable cells expressing AirID-IκBα, RelA biotinylation was clearly found by supplementing with >50 µM for 6 hr. It was not found in cells in which AirID was stably expressed (*Figure 4—figure*

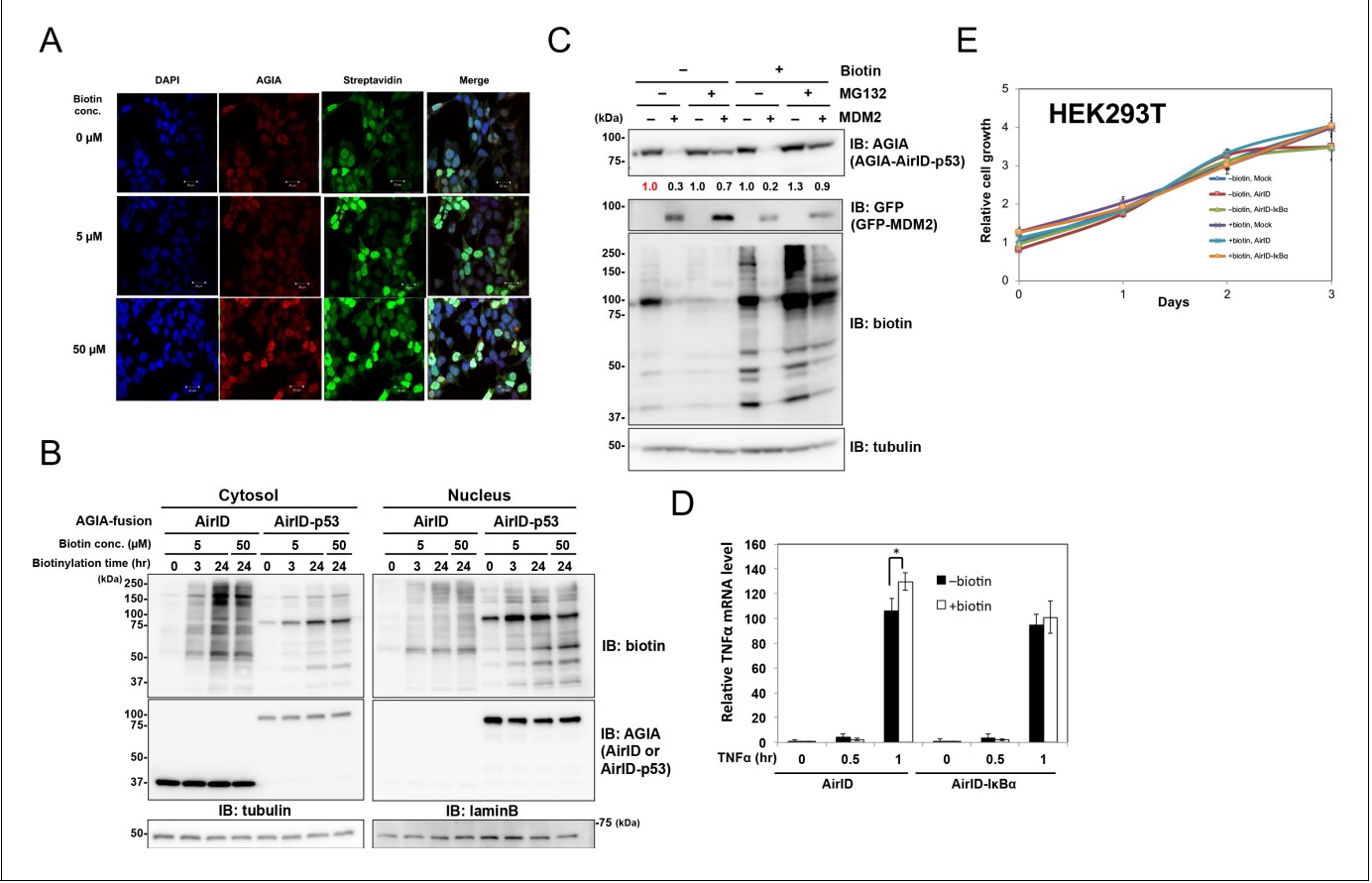

**Figure 4.** Dynamics and effects of AirID- and AirID-fusion proteins in cells. Localization analysis was carried out for AirID using (**A**) immunostaining and (**B**) a fractionation assay. For immunostaining, HEK293 cells overexpressing AGIA-AirID-p53 were supplemented with the described biotin concentration for 3 hr. The cells were immobilized using anti-AGIA antibody and visualized using anti-rabbit IgG antibody-AlexaFluor555 and streptavidin-AlexaFluor488. AGIA-tagged AirID or AirID-p53 was transfected in HEK293T for the fractionation assay. The next day, biotin was added to 5 µM or 50 µM for the described time. Cytoplasmic and nuclear proteins were fractionated using a ProteoExtract Subcellular Proteome Extraction kit (Merck). As a control, expression of AirID or AirID-p53 was detected using anti-AGIA antibody. (**C**) AGIA tagged AirID-p53 was co-transfected with or without GFP-MDM2 in HEK293T. Biotin was added to a concentration of 50 µM at the same time. After 6 hr, DMSO or MG132 was added to a concentration of 10 µM. As a control, expression of MDM2 was detected using anti-GFP antibody. The band intensity of AirID-p53 was quantified with image J software. The index intensity (value 1.0) is shown in red characters. (**D**) qRT-PCR using AirID-IκBα. AGIA-tagged AirID or AirID-IκBα was stably expressed using renti-virus in HEK293T. Cells were seeded in a 96-well plate, and biotin was added at the same time. Next day, cells were stimulated using TNFα (20 ng/mL) for 0, 0.5, or 1 hr. In the cells, the mRNA level of TNFα was analyzed by qRT-PCR. Mean ± S.D. (n = 3). *, p<0.05. (**E**) Viability of AirID-expressing cells. AGIA-tagged AirID or AirID-IκBα was stably expressed using renti-virus in HEK293T. Cells were seeded in 96-well plates, and biotin was added the next day. The MTS assay was performed 0, 1, 2, or 3 days after adding biotin to measure cell viability.

The online version of this article includes the following source data and figure supplement(s) for figure 4:

**Source data 1.** Cell growth analysis data relating to *Figure 4E*.
**Source data 2.** qRT-PCR data related to *Figure 4D*.
**Figure supplement 1.** Biotinylation conditions of cells stably expressing AirID.
**Figure supplement 2.** Comparison of the effect on cell growth of TurboID and AirID.
**Figure supplement 2—source data 1.** Viability analysis data related to *Figure 4—figure supplement 2*.

*supplement 1*). The cell growth of both cell types was not fully inhibited by 50 µM biotin supplementation (*Figure 4E*), indicating that AirID does not affect cell viability under 50 µM biotin supplementation conditions. It was demonstrated that TurboID showed cytotoxicity within 48 hr with 50 µM biotin (*Branon et al., 2018*). Therefore, AirID- or TurboID-expressing cells were cultured with or without biotin for 48 hr and then the viability of them was analyzed. Compared with control (Mock), the viability of TurboID-expressing cell was significantly decreased with 50 µM biotin, but the viability of AirID-expressing cells was not significantly affected (*Figure 4—figure supplement 2*).

## Biotinylation of endogenous proteins by AirID in cells

We investigated whether AirID could biotinylate dependently interacting endogenous proteins in cells. Streptavidin-conjugated beads were used as in a previous report to recover biotinylated endogenous proteins from cell lysates (*Van Itallie et al., 2013*). AGIA-AirID-IκBα was transiently expressed in HEK293T cells under different biotin concentration conditions. Streptavidin-pull down assay of the cell lysates was carried out, and the biotinylated endogenous proteins were detected using immunoblotting with each specific antibody. Endogenous RelA protein was biotinylated without supplemental biotin by transiently expressing AGIA-AirID-IκBα in cells (IB: RelA, 0 µM biotin concentration in *Figure 5A*). Biotin supplementation enhanced biotinylations of p50 or p105, which are known IκBα interactors (IB: p50/p105 and p50, 5 µM or 10 µM biotin concentration). These biotinylations were not found for AGIA-AirID alone. As the IκBα protein interacts with RelA, this result illustrated that biotinylated AGIA-AirID-IκBα may bring endogenous RelA without biotinylation. To confirm this, immunoprecipitation using a specific antibody recognizing endogenous RelA was performed in severe conditions after proteins were denatured with 1% SDS. Biotinylation of endogenous RelA recovered by immunoprecipitation was observed as a result (*Figure 5B*). Using the same lysates, the streptavidin-pull down assay recovered RelA protein, indicating that RelA biotinylation depends on AGIA-AirID-IκBα. These results showed IκBα-interaction-dependent biotinylation by AirID.

As in vitro protein biotinylation involving in a CRL4$^{AirID-CRBN}$ complex was found (*Figure 3E*), we investigated whether proteins in the CRL4 complex were biotinylated by AirID-fusion CRBN in cells. AGIA-AirID-CRBN was transiently expressed in HEK293T cells, and cell lysates were pulled down using streptavidin beads. Biotinylation of CUL4 and RBX1 was found after supplementing with 5 µM biotin (*Figure 5C*), but DDB1 biotinylation was not found. Taken together, these results indicate that the biotinylation assay using the AirID-fusion target is a useful tool for analyzing PPI in the cell.

## Mass spectrometry analysis of AirID-IκBα-dependent biotinylated proteins in cells

Since BioID has been widely used to identify PPI in cells using mass spectrometry (MS) analysis (*Ikeda and Freeman, 2019*), we also analyzed biotinylated proteins using LC-MS/MS in the cells stably expressing AirID alone or AirID-IκBα that were used in *Figure 4E*. The flowchart for the analysis of biotinylated peptides is shown in *Figure 6A*. The cells were treated with 50 µM biotin for 6 hr. Proteins were digested using trypsin after cell lysis. Biotinylated peptides were captured using Tamavidin2-Rev. Tamavidin2-Rev can bind to biotin-labelled substances and can release them under high concentrations of free biotin (*Takakura et al., 2013*). The biotinylated peptides were eluted using 2 mM biotin, and the eluted peptides were analyzed using LC-MS/MS. The biotinylation by free biotinoyl-5′-AMP occurs on lysine (Lys) residues on proteins (*Choi-Rhee et al., 2004*). Trypsin digests Lys or arginine (Arg), but it cannot cleave modified Lys in the same way as it does biotinylated Lys (*Bheda et al., 2012*). These features show that an eluted biotinylated peptide has a single biotin, indicating that the direct determination of biotinylated peptide provides a biotinylation site on the peptide. Using this method, we found 12 biotinylated peptides that were present in AirID-IκBα-expressing cells at levels that were more than five times higher than those in cells expressing only AirID (*Figure 6B and C*, *Figure 6—source data 1*). In the top five peptides, three biotinylated peptides were derived from RelA proteins (*Figure 6C*), indicating that AirID-IκBα could accurately biotinylate a major partner RelA protein in the cells. Furthermore, we investigated whether AirID-dependent biotinylation occurs in a specific region. Comparison of amino-acid sequences among the top 20 biotinylated peptides showed no similarity except for a single Lys residue (*Figure 6—figure supplement 1*), suggesting that the proximate biotinylation by AirID happens on the Lys residue but does not have a preferred sequence.

In *Figure 5A*, the streptavidin pull down clearly showed biotinylation of the endogenous RelA protein in transiently AirID-IκBα expressing cells. We assessed whether an AirID-IκBα-dependent biotinylation of RelA could be detected using LC-MS/MS in transiently AirID-IκBα expressing cells. A flowchart for the analysis of biotinylated peptides using transiently expressing cells is shown in *Figure 6D*. As expected, the top biotinylated peptide was RelA (*Figure 6E*), as in stably expressing cells. Taken together, these results suggest that detection of AirID-dependent biotinylation using LC-MS/MS is useful for PPI analysis in cells.

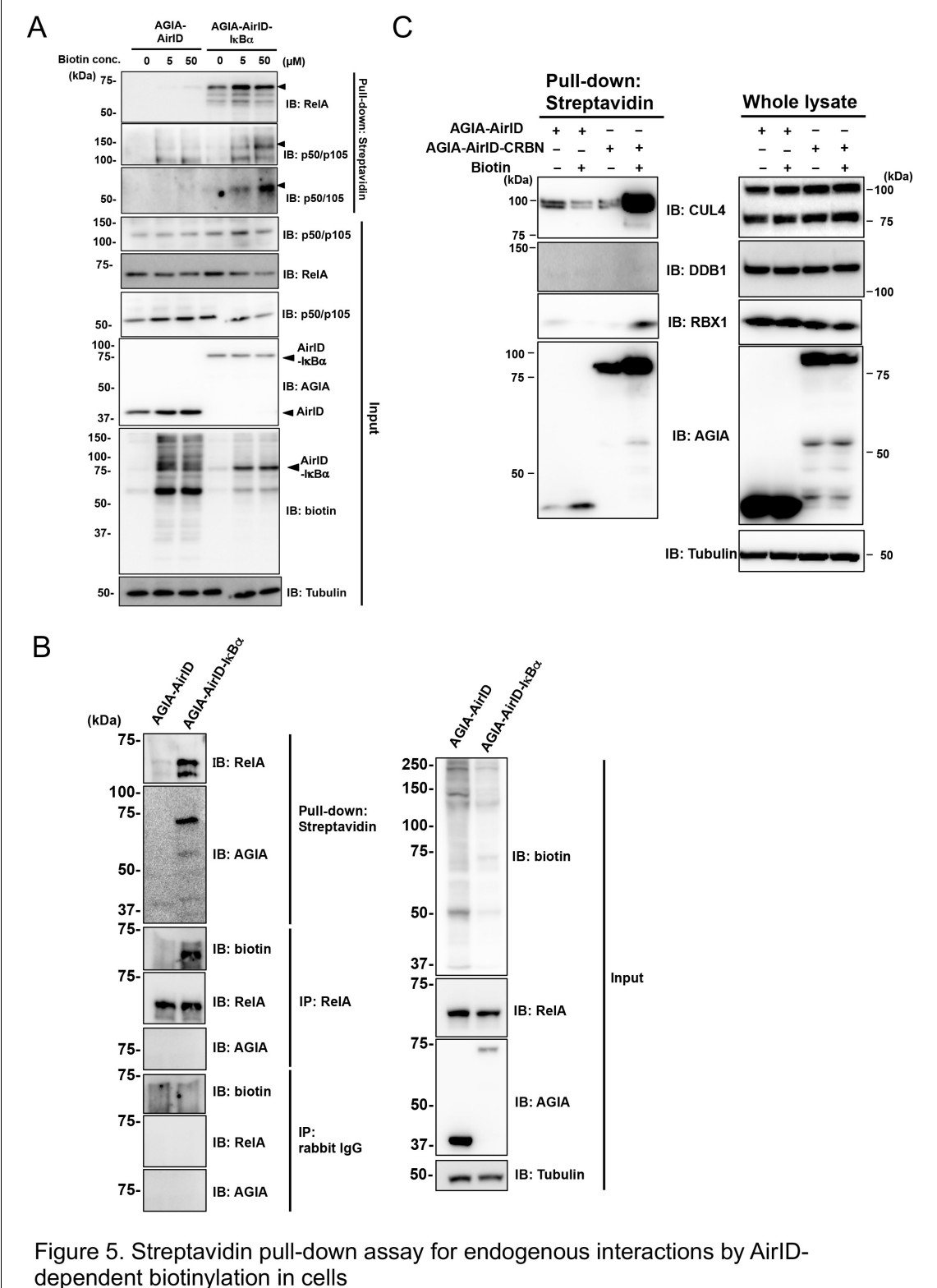

Figure 5. Streptavidin pull-down assay for endogenous interactions by AirID-dependent biotinylation in cells

**Figure 5.** Streptavidin pull-down assay for endogenous interactions using AirID-dependent biotinylation in cells. (**A**) Biotin was added to HEK293T cells expressing AirID or AirID-IκBα to concentrations of 0, 5, or 50 μM before incubating for 3 hr. Cells were lysed before immunoprecipitating with streptavidin beads. Pulled-down proteins were detected using immunoblotting with the described antibody. (**B**) AirID- or AirID-IκBα-expressing HEK293T cells were supplemented with 5 μM biotin and incubated for 3 hr. Cells were lysed and pulled down with streptavidin beads or

*Figure 5 continued on next page*

*Figure 5 continued*

immunoprecipitated with anti-RelA. Normal rabbit-IgG was also used as a negative immunoprecipitation control. Pulled-down or immunoprecipitated proteins were detected using immunoblotting with the described antibody. (**C**) Biotinylation of the CRL4$^{CRBN}$ complex was performed using AirID-CRBN. AirID- or AirID-CRBN-expressing HEK293T cells were incubated with or without 5 μM of biotin for 3 hr. Cells were lysed and pulled down with streptavidin. CUL4, DDB1, and RBX1 were detected using immunoblotting with each antibody.

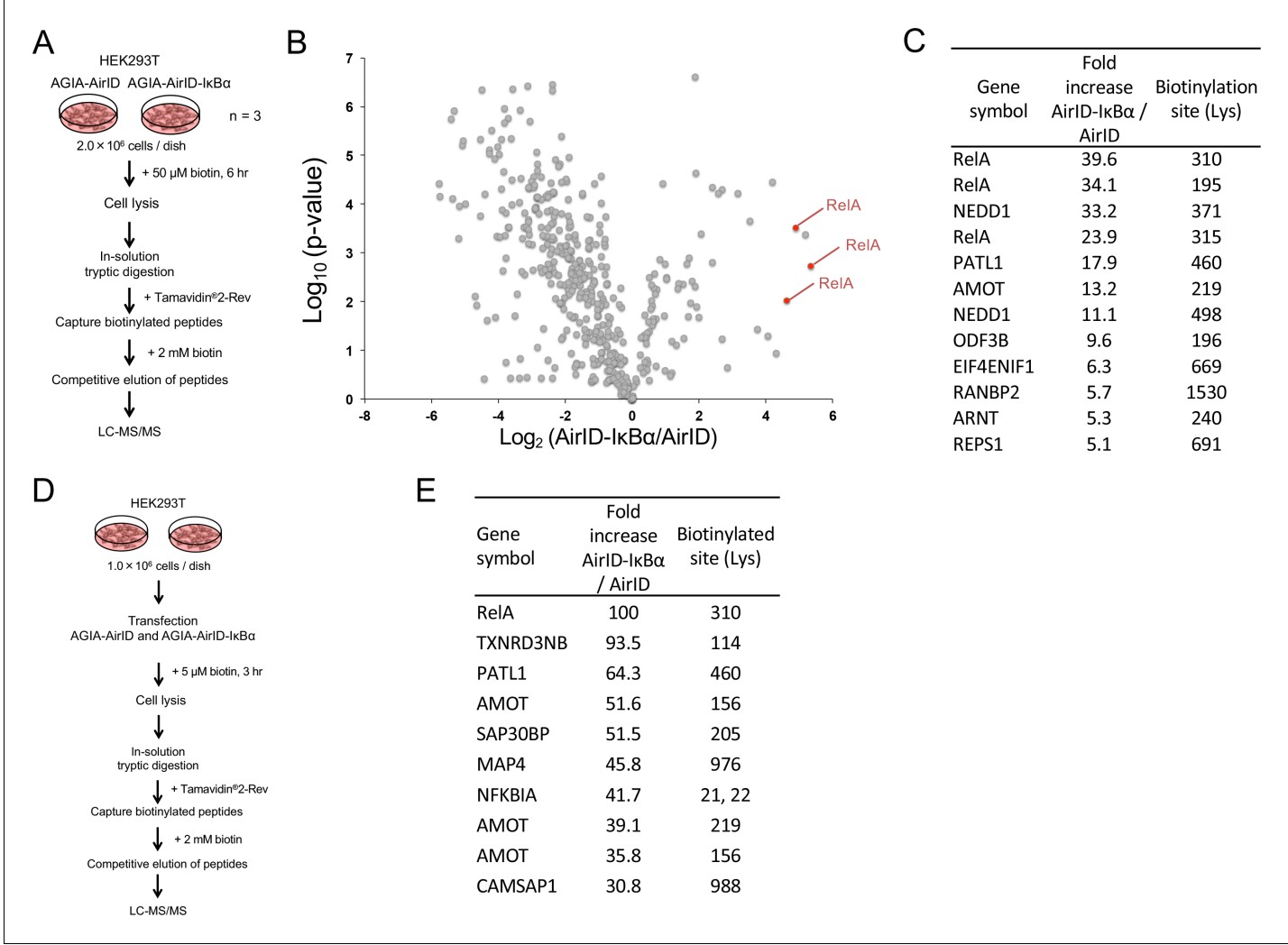

**Figure 6.** Mass spectrometry analysis of biotinylated proteins in AirID-IκBα expressing cells. (**A**) Schematic figure for detecting biotinylated proteins using cells stably expressing AirID. HEK293T cells that stably expressed AGIA-AirID or AGIA-AirID-IκBα were cultured in DMEM containing 50 μM for 6 hr before collecting (n = 3). Collected cells were lysed, and proteins were digested in solution using trypsin. Biotinylated peptides were captured from digested peptides using Tamavidin2-Rev beads (Wako), which can elute biotinylated samples using 2 mM biotin. Eluted peptides were detected using LC-MS/MS. (**B**) A volcano plot showing AirID-IκBα versus AirID against p-value of triplicate experiments. (**C**) A list of peptides increased by more than 5-fold. (**D**) Schematic figure for detecting biotinylated proteins using cells transiently expressing AirID. HEK293T cells that transiently expressed AGIA-AirID or AGIA-AirID-IκBα were cultured in DMEM containing 5 μM for 3 hr before collecting (n = 1). Biotinylated proteins were detected using a similar method. (**E**) A list of the top ten peptides increased by AirID-IκBα.

The online version of this article includes the following source data and figure supplement(s) for figure 6:

**Source data 1.** Mass spectrometry data related to *Figure 6B*.
**Figure supplement 1.** Biotinylation site characteristics.

## Discussion

Here, we used an algorithm of ancestral enzyme reconstruction using a large genome dataset, and we investigated five ancestral BirA enzymes. Finally, we combined biochemical experiments and RS mutations to create AirID with high PPI proximity biotinylation. Classical evolutionary protein engineering used random mutations to improve the activity (*Branon et al., 2018*). Therefore, the sequence similarity is extremely high because random mutations cannot provide dynamic sequence changes. However, sequence similarity between *E. coli* BirA and ancestral BirA was between 40% to 80%, indicating that a computational approach using large genome datasets can more dynamically design enzyme sequences. As another aspect to this approach, the BirA active region ($_{115}$GRGRRG$_{121}$) (*Kwon and Beckett, 2000*) was conserved, and RG and RS mutations introduced into ancestral BirA enzymes (*Figures 1* and *2*). In the present direction of computational protein evolution, dynamic changes to the backbone region of protein enzymes with a conserved active pocket would be acceptable. Further accumulation of knowledge about the enzyme function would be required to change the enzyme active region dynamically.

When we looked at BioID (BirA*), TurboID, and AirID, the proximal biotinylation activity of BioID (BirA*) was considerably lower than that of TurboID and AirID (*Figure 2A* and *Figure 2—figure supplement 1*). By contrast, TurboID showed the highest proximate biotinylation activity in vitro and in cells (*Figure 2A* and *Figure 2—figure supplement 1*). This enzyme could be used for biotinylation within one hour (*Branon et al., 2018*). However, the highest activity from TurboID provided extra biotinylation on unexpected proteins, such as like GFP or tubulin, in cells that were treated for a long incubation of more than six hours and higher biotin concentrations (such as 50 μM biotin) (*Figure 2D*). In the first report describing TurboID, it was used as a biotin-labelling enzyme rather than as a proximal biotinylation enzyme for PPI (*Branon et al., 2018*). If it was used analyze PPI, TurboID would show the best performance under limiting conditions, such as a short treatment (1 hr) in cells.

In the case of AirID, GFP and tubulin biotinylations were not observed in the same conditions as those catalyzed by TurboID (*Figure 2D*). Streptavidin-pull down assay and LC-MS/MS analysis also indicated that AirID-fusion proteins were able to biotinylate each well-known interactor accurately in the transient- and stable-expression cells (*Figures 5* and *6*). The formation of biotinoyl-5′-AMP was greater for AirID than for TurboID in low ATP concentrations (1 μM) (*Figure 2—figure supplement 4C*), and it prefers lower concentrations of biotin (with 5 μM biotin or without biotin supplement) (*Figure 2—figure supplement 2*). In addition, analysis of biotinylation sites from LC-MS/MS showed that AirID biotinylation happened with no special sequence preference on a proximate Lys residue (*Figure 6—figure supplement 1*). Taken together, our AirID is expected to enhance PPI-dependent biotinylation accuracy, suggesting that AirID is suitable for PPI analysis in cells.

Inhibition of MDM2–p53 interaction by nutline-3 was detected using AirID biotinylation (*Figure 3A and C*), and several pomalidomide-dependent interactions between CRBN and neosubstrates were also detected by AirID biotinylation (*Figure 3D*). These results indicate that AirID-dependent biotinylation would be useful for PPI analysis using chemical compounds. Furthermore, in vivo proximity biotinylation using BioID has been performed in many studies because the identification of in vivo partner proteins of target proteins is key for understanding biological functions (*Odeh et al., 2018*; *Motani and Kosako, 2018*), and it has uncovered new PPIs. Stable expression of AirID-IκBα did not induce cell-growth inhibition even under biotin-supplementation conditions (*Figure 4E*), suggesting that AirID-fusion protein expression would have very low toxicity. Therefore, AirID could also be used for in vivo screening for protein interactors of a target protein. In conclusion, AirID is a novel enzyme providing proximity biotinylation for PPI analysis.

## Materials and methods

**Key resources table**

| Reagent type (species) or resource | Designation | Source or reference | Identifiers | Additional information |
|---|---|---|---|---|
| Gene (*E. coli*-modified) | BioID | *Kim et al., 2014*; DOI: 10.1073/pnas.1406459111 | | Obtained by mutating 118Arg of BirA to Gly |
| *Continued on next page* | | | | |

*Continued*

| Reagent type (species) or resource | Designation | Source or reference | Identifiers | Additional information |
|---|---|---|---|---|
| Gene (*E. coli*-modified) | TurboID | *Branon et al., 2018* DOI: 10.1038/nbt.4201 | | Synthetic gene fragment was purchased using Invitrogen Gene Art |
| Gene (artificially designed) | AVVA | This paper | | Synthetic gene fragment was purchased using Invitrogen Gene Art |
| Gene (artificially designed) | AFVA | This paper | | Synthetic gene fragment was purchased using Invitrogen Gene Art |
| Gene (artificially designed) | AHLA | This paper | | Synthetic gene fragment was purchased using Invitrogen Gene Art |
| Gene (artificially designed) | GFVA | This paper | | Synthetic gene fragment was purchased using Invitrogen Gene Art |
| Gene (artificially designed) | All | This paper | | Synthetic gene fragment was purchased using Invitrogen Gene Art |
| Gene (*Homo sapiens*) | IκBα | Mammalian Gene Collection | | |
| Gene (*H. sapiens*) | RelA | Mammalian Gene Collection | | |
| Gene (*H. sapiens*) | p53 | Mammalian Gene Collection | | |
| Gene (*H. sapiens*) | Mdm2 | Mammalian Gene Collection | | |
| Gene (*H. sapiens*) | CRBN | Mammalian Gene Collection | | |
| Gene (*H. sapiens*) | RBX1 | Mammalian Gene Collection | | |
| Gene (*H. sapiens*) | DDB1 | Mammalian Gene Collection | | |
| Gene (*H. sapiens*) | CUL4 | Mammalian Gene Collection | | |
| Gene (*H. sapiens*) | SALL4 | Mammalian Gene Collection | | |
| Gene (*H. sapiens*) | IKZF1 | Mammalian Gene Collection | | |
| Recombinant DNA reagent | pEU (plasmid) | Cell-Free Science | | For protein expression using wheat germ cell-free system |
| Recombinant DNA reagent | pcDNA3.1 (plasmid) | Modified Invitrogen | | For protein expression using human cell |
| Recombinant DNA reagent | pET30a (plasmid) | Modified Merck | | For protein expression using *E. coli* |
| Antibody | Anti-AGIA HRP | *Yano et al., 2016* DOI: 10.1371/journal.pone.0156716 | Rabbit mAb | WB (1:10,000) |
| Antibody | Anti-biotin, HRP-linked antibody (from goat) | Cell Signaling | #7075S RRID:AB_10696897 | WB (1:10,000) |

*Continued on next page*

*Continued*

| Reagent type (species) or resource | Designation | Source or reference | Identifiers | Additional information |
|---|---|---|---|---|
| Antibody | His-probe antibody (H-3) (mouse mAb) | SantaCruz | sc-8036 RRID:AB_627727 | WB (1:1000) |
| Antibody | Anti-FLAG M2-HRP (mouse mAb) | Sigma | A8592 RRID:AB_439702 | WB (1:10,000) |
| Antibody | Anti-GST-tag pAb-HRP-DirecT (rabbit pAb) | MBL | PM013-7 RRID:AB_10598029 | WB (1:1000) |
| Antibody | Anti-p65 (D14E12) (rabbit mAb) | Cell Signaling | #8242S RRID:AB_10859369 | WB (1:1000) |
| Antibody | Anti-GFP (1E4) (mouse mAb) | MBL | M048-3 RRID:AB_591823 | WB (1:5000) |
| Antibody | Anti-α-tubulin pAb-HRP-DirecT (rabbit pAb) | MBL | PM054-7 RRID:AB_10695326 | WB (1:10,000) |
| Antibody | Anti-laminB (goat pAb) | SantaCruz | sc-6217 RRID:AB_648158 | WB (1:1000) |
| Antibody | Anti-Myc (4A6) (mouse mAb) | Merck | 05–724 RRID:AB_11211891 | WB (1:2000) |
| Antibody | Anti-p50/p105 (D7H5M) (rabbit mAb) | Cell Signaling | #12540S RRID:AB_2687614 | WB (1:1000) |
| Antibody | Anti-CUL4 (H-11) | SantaCruz | sc-377188 | 1:1000; 5% skimmed milk in TBST |
| Antibody | Anti-DDB1 (E-11) | SantaCruz | sc-376860 | 1:1000; 5% skimmed milk in TBST |
| Antibody | Anti-RBX1 (E-11) | SantaCruz | sc-393640 RRID:AB_2722527 | 1:1000; 5% skimmed milk in TBST |
| Antibody | Anti-Rabbit IgG, HRP-Linked F (ab')$_2$ Fragment Donkey | GE Healthcare | NA9340V RRID:AB_772191 | WB (1:10,000) |
| Antibody | Anti-Mouse IgG, HRP-Linked F (ab')$_2$ Fragment Sheep | GE Healthcare | NA9310V RRID:AB_772193 | WB (1:10,000) |
| Antibody | Normal rabbit IgG | MBL | PM035 | |
| Antibody | Streptavidin, Alexa Fluor 488 | Thermo | S32354 RRID:AB_2315383 | |
| Antibody | F(ab')$_2$-Goat anti-Rabbit IgG (H+L) Cross-Adsorbed Secondary Antibody, Alexa Fluor 555 | Thermo | A21431 RRID:AB_1500601 | |
| Cell line (*H. sapiens*) | HEK293T | RIKEN BRC | RCB2202 | Maintained in DMEM supplemented with 10% FBS |
| Beads | Dynabeads proteinG | Invitrogen | DB10004 | |
| Beads | Ni Sepharose High Performance | GE Healthcare | 17526801 | |
| Beads | Glutathione Sepharose 4B | GE Healthcare | 17526801 | |
| Beads | Streptavidin Sepharose High Performance | GE Healthcare | 17511301 | |
| Beads | Tamavidin 2-Rev magnetic beads | Fujifilm | 133–18611 | |
| Commercial assay kit | Alphascreen Protein A detection kit | Perkin Elmer | 6760617R | |

*Continued on next page*

*Continued*

| Reagent type (species) or resource | Designation | Source or reference | Identifiers | Additional information |
| --- | --- | --- | --- | --- |
| Commercial assay kit | CellTiter 96 AQueous One Solution Cell Proliferation Assay | Promega | G5430 | |
| Commercial assay kit | SuperPrep II Cell Lysis and RT kit for qPCR | TOYOBO | SCQ401 | |
| Commercial assay kit | KOD SYBR qPCR Mix | TOYOBO | QKD-201 | |
| Commercial assay kit | Wheat germ cell-free protein synthesis kit | Cell-Free Science | | |
| Chemical compound and drug | Fetal Bovine Serum | Wako | | |
| Chemical compound and drug | DMEM, low glucose | Wako | 041–29775 | |
| Chemical compound and drug | Nutlin-3 | Sigma | N6287-1MG | |
| Chemical compound and drug | Pomalidomide | TCI | P2074 | |
| Chemical compound and drug | MG-132 | Peptide Institute | 3175 v | |
| Chemical compound and drug | D-Biotin | Nacalai tesque | 04822–91 | |
| Chemical compound and drug | Protease inhibitor cocktail | Sigma | P8340-5ML | |
| Chemical compound and drug | Penicillin-Streptomycin | Thermo | 15140122 | |
| Chemical compound and drug | Isobutyric acid | Nacalai tesque | 06429–85 | |
| Chemical compound and drug | Ammonia solution | Nacalai tesque | 025–12 | |
| Other | cellulose TLC plate | Merck | 1.05552.0001 | Toll for chromatography |

## Reconstruction of five ancestral BirA

The BirA homologous sequences, classified into four groups using the key residues, were aligned using MAFFT software 2 (*Katoh et al., 2002*). Each of the aligned sequences was analyzed using MEGA6 software 3, and the phylogenetic tree was generated using the maximum likelihood method (*Tamura et al., 2013*). Aligned sequences and phylogenetic tree data were submitted to FastML (*Ashkenazy et al., 2012*). The JTT empirical model was adopted for analysis. Finally, we obtained four ancestral BirA forms named AVVA, AFVA, AHLA, and GFVA. Furthermore, we applied the three designed sequences (AVVA, AFVA, and GFVA) and an identical procedure to design another ancestral BirA called 'all'. All of the five designed sequences are shown in *Supplementary file 1*.

## Plasmids

BioID (BirA*), TurboID, or ancestral BirAs (AncBirAs) were cloned into pcDNA3.1-AGIA or pEU-AGIA vectors using the BamH1 and Not1 restriction sites. AncBirAs-fused p53, IκBα or CRBN were cloned into pcDNA3.1-AGIA or pEU-AGIA vectors using BamH1, Kpn1, and Not1. Mutants of AncBirAs were generated using site-directed mutagenesis with a PrimeSTAR mutagenesis basal kit (Takara). MDM2 and RelA were cloned into pEU-FLAG-GST or pcDNA6.2-GFP vectors using the Gateway cloning system (Thermo Fisher Scientific). Lentivirus-based AGIA-tagged AirID and AirID-IκBα plasmids were generated using restriction enzyme digestion of a CS II-CMV-MCS-IRES2-Bsd vector. For *E. coli* expression, TurboID, GFVA-R118G, or AirID was cloned into the pET30a-His vector using an In-Fusion HD Cloning Kit (Takara).

## Cell lines

HEK293T cells (purchased from RIKEN RCB, Tsukuba, Japan, catalog number RCB2202) were incubated at 37 °C and 5% $CO_2$ in Dulbecco's Modified Eagle Medium (DMEM) (wako) supplemented with 10% fetal bovine serum (Biosera) and antibiotics (100 units/mL penicillin and 100 µg/mL streptomycin) (Thermo). We confirmed that the cell line was free of mycoplasma contamination. Lentiviruses expressing AGIA-AirID and AGIA-AirID-IκBα were generated by transfection using PEI MAX - Transfection Grade Linear Polyethylenimine Hydrochloride (Polyscience). After transmission of the transgene, a pool of HEK293T cells that were resistant to Blasticidin S (10 µg/mL) (Invitrogen) was generated and used in subsequent experiments.

## Cell-free protein synthesis and GST-tag purification

In vitro transcription and wheat cell-free protein synthesis were performed using the WEPRO1240 expression kit (Cell-Free Sciences). A transcript was made from each of the DNA templates mentioned above using SP6 RNA polymerase. The translation reaction was performed using the WEPRO1240 expression kit (Cell-Free Sciences). For biotin labelling, 1 µl of BirA or of the ancestral BirAs produced by the wheat cell-free expression system were added to the bottom layer, and 500 nM (final concentration) of D-biotin (Nacalai Tesque) was added to both upper and bottom layers as described previously (*Sawasaki et al., 2008*). The aliquots were used for expression analysis and functional characterization. 1 mL of synthesized His-bls-FLAG-GST was mixed with Glutathione Sepharose 4B (GE Healthcare) and rotated for 3 hr at 4 °C. The mixture was washed with PBS. Proteins were eluted in 100 µL fractions with elution buffer (50 mM Tris-HCl [pH8.0], 10 mM reduced glutathione). Protein was subjected to SDS-PAGE and CBB staining to determine purity.

## BirA enzyme preparation from *E. coli*

To purify TurboID, GFVA-R118G, and AirID proteins, the genes encoding them were inserted into pET30a and transformed into *E. coli* strain BL21. The *E. coli* cells were grown at 37 °C in LB medium to an OD600 of 0.6 and induced by adding IPTG to 1 mM for an additional 6 hr at 37 °C. Cells were centrifuged and resuspended in lysis buffer (20 mM sodium phosphate, 300 mM NaCl, 10 mM imidazole). The cells were lysed using sonication, and the lysates were centrifuged. The supernatants were added to Ni Sepharose High Performance (GE Healthcare) and incubated for 3 hr at 4 °C. The mixture was washed with three column volumes of wash buffer (20 mM sodium phosphate, 300 mM NaCl, 50 mM imidazole). Proteins were eluted in 500 µL fractions with elution buffer (20 mM sodium phosphate, 300 mM NaCl, 500 mM imidazole). Fractions were dialyzed against PBS. Proteins were subjected to SDS-PAGE and CBB staining to determine purity.

## Cell transfection and immunoblotting

HEK293T cells were transfected with various plasmids using PEI MAX (Polyscience). Immunoblotting was performed according to standard protocols. Briefly, proteins in whole-cell lysates were separated using SDS-polyacrylamide gel electrophoresis (SDS-PAGE) and transferred onto a PVDF membrane using semi-dry blotting. After blocking with 5% milk/TBST or Blocking one (Nakalai Tesque), the membrane was incubated with the appropriate primary antibodies followed by a horseradish peroxidase (HRP)-conjugated secondary antibody.

## Biotinylation assays

In vitro biotinylation assays were performed. Briefly, 5 µL of each synthesized protein was mixed and incubated at 26 °C for 1 hr. Biotin was added, and the biotinylation reaction was performed in a total volume of 15 µL. After the reaction, biotinylated proteins were analyzed using SDS-PAGE and immunoblotting. In cell biotinylation assays were also performed. Briefly, each BirA or BirA fused gene and substrate gene were transfected into HEK293T. At the same or each time, biotin was added and cells were lysed using SDS sample buffer (125 mM Tris-HCl [pH 6.8], 4% SDS, 20% glycerol, 0.01% BPB, 10% 2-mercaptoethanol) 24 hr after transfection. Whole cell lysates were analyzed using SDS-PAGE and immunoblotting.

## Streptavidin pull-down assays

A streptavidin pull-down was performed to analyze biotinylated proteins. In vitro reaction mixtures after biotinylation were diluted with wash buffer (50 mM Tris-HCl [pH7.5], 1% TritonX-100, 150 mM NaCl), and 1% SDS was added to stop the reaction. After biotinylation, cells were lysed using RIPA buffer (50 mM Tris-HCl [pH 8.0], 150 mM NaCl, 0.5% sodium deoxycholate, 0.1% SDS, 1% NP-40) including protease inhibitor cocktail and sonication. Lysates were centrifuged at 15,000 rpm at 4 °C for 10 min. SDS was added to supernatants to 1%. In vitro reaction mixtures or cell lysates were mixed with equilibrated Streptavidin Sepharose High Performance (GE Healthcare) and rotated at 4 °C for 1 hr. After flow-through was removed, beads were washed three times using wash buffer (50 mM Tris-HCl [pH7.5], 1% TritonX-100, 150 mM NaCl), and beads were boiled into SDS sample buffer. Boiled solution was analyzed using SDS-PAGE and immunoblotting.

## In vitro inhibition assays using AlphaScreen technology

Synthesized FG-MDM2 and Nutlin-3 were mixed and incubated for 30 min at 26 °C. AGIA-AirID-p53 was added to the mixture and incubated for 1 hr at 26 °C. In addition, biotin was added to the reaction mixture to 500 nM and incubated for 3 hr at 26 °C. Inhibition was examined using the AlphaScreen IgG (Protein A) detection kit (Perkin Elmer) and immunoblotting. Briefly, for AlphaScreen, 10 µL of detection mixture containing 100 mM Tris-HCl (pH 8.0), 0.1% Tween 20, 100 mM NaCl, 10 ng anti-FLAG antibody (Sigma), 1 mg/mL BSA, 0.1 µL streptavidin-coated donor beads, and 0.1 µL protein A-conjugated acceptor beads were added to each well of a 384-well Opti-plate before incubation at 26 °C for 1 hr. Luminescence was detected using the AlphaScreen detection program with an EnVision device (PerkinElmer). For immunoblotting, solutions were boiled in SDS sample buffer. The boiled solution was analyzed using SDS-PAGE and immunoblotting.

## Immunoprecipitation

Cells after biotinylation were lysed with RIPA buffer and sonication for immunoprecipitation. Lysates were centrifuged and SDS was added to supernatants to denature proteins. Their solutions were diluted 10-fold. After 2 µg of the indicated antibodies were bound to either protein A or protein G Dynabeads (Thermo Fisher Scientific) for 30 min at room temperature, the beads were incubated with cell lysates diluted overnight at 4 °C. The immunocomplexes were boiled in SDS sample buffer after washing three times with PBS. The boiled solution was analyzed using SDS-PAGE and immunoblotting.

## Cell viability assays

Cells were seeded into 96-well plates at a density of $0.25 \times 10^4$ cells/well and treated with 50 µM biotin after 24 hr. Cell viability was determined using the MTS assay with a CellTiter96 Aqueous One Solution Cell Proliferation Assay kit (Promega). In brief, 20 µL of the MTS reagent was added into each well, and the cells were incubated at 37 °C for 1 hr. The absorbance was detected at 490 nm (reference: 650 nm) with a Microplate Reader (SpectraMaxM3 Multi-Mode Microplate Reader; Molecular Devices).

Cells were seeded into 96-well plates at a density of $0.25 \times 10^4$ cells/well and transfected after 24 hr. After 2 days, cell viability was determined using CellTiter-Glo Luminescent Cell Viability Assay Cell system (Promega). In brief, the CellTiter-Glo reagent was added into each well, and the cells were incubated at room temperature for 10 min. The luminescence was detected with a Microplate Reader (GloMax Discover Microplate Reader).

## Biotinoyl-5′-AMP synthesis assay

The assays contained 50 mM Tris-HCl buffer (pH 8.0), 5.5 mM $MgCl_2$, 100 mM KCl, 0.1 mM TCEP, 1 µM ATP including [$\alpha$-$^{32}$P]ATP, 25 µM biotin, 2 µM BirA and with or without 1 µM His-bls-FLAG-GST for a total reaction mixture of 10 µl (*Henke and Cronan, 2014*). The reaction mixtures were incubated at 37 °C for 30 min. A portion of each reaction mixture (1 µl) was spotted onto cellulose thin-layer chromatography (TLC) plates and developed in isobutyric acid-$NH_4OH$-water (66:1:33) (*Prakash and Eisenberg, 1979*). The thin-layer chromatograms were exposed to a phosphorimaging screen and visualized using a Typhoon FLA-3000 (GE Healthcare).

## Fractionation assay

HEK293T cells were seeded onto a 24-well plate. Next day, cells were transfected, and biotin was added at the same time or each time. Subcellular fractionation was performed 24 hr after transfection using a ProteoExtract Subcellular Proteome Extraction kit (Merck) according to the protocol.

## Immunofluorescent staining

Cells were fixed with 4% paraformaldehyde in phosphate-buffered saline (PBS) for 15 min at room temperature before permeabilizing with 0.5% Triton X-100 in PBS for 15 min. Cells were incubated with a primary antibody overnight at 4 ˚C after blocking with 0.5% CS in TBST for 1 hr. After washing with TBST, cells were incubated with the appropriate Alexa Flour 488- and/or 555-conjugated secondary antibody and streptavidin for 1 hr at room temperature. Nuclei were counterstained with 4,6-diamidino-2-phenylindole. After washing with TBST, coverslips were mounted with anti-fade.

## Quantitative RT-PCR

Preparation of cell lysates and reverse transcription were performed using SuperPrep II Cell Lysis and RT Kit for qPCR (TOYOBO). Real-time PCR was carried out using KOD SYBR qPCR Mix (TOYOBO). qRT-PCR primers used were as follows; TNF-α: 5′- CAGCCTCTTCTCCTTCCTGAT (forward), 5′-GCCAGAGGGCTGATTAGAGAGA (reverse); GAPDH: 5′-AGCAACAGGGTGGTGGAC (forward), 5′- GTGTGGTGGGGGACTGAG (reverse).

## Mass spectrometry analysis of biotinylated peptides

The proximity-dependent biotin identification method using AirID was performed according to a previous report (*Kim et al., 2016*). Briefly, confluent HEK293T cells stably expressing AirID or AirID-IκBα fused at the N-terminus with an AGIA tag in a 6 cm dish were incubated with 50 μM biotin for 6 hr before harvesting using ice-cold PBS. Cell pellets were lysed and digested with trypsin. The digested peptides were incubated with Tamavidin2-Rev magnetic beads (FUJIFILM) before eluting with 2 mM biotin. Detailed procedures will be described elsewhere (Motani K and Kosako H, in preparation).

LC-MS/MS analysis of the resulting peptides was performed on an EASY-nLC 1200 UHPLC connected to a Q Exactive Plus mass spectrometer using a nanoelectrospray ion source (Thermo Fisher Scientific). The peptides were separated on a 75-μm inner diameter ×150 mm C18 reverse-phase column (Nikkyo Technos) with a linear gradient from 4–28% acetonitrile for 0–40 min followed by an increase to 80% acetonitrile during 40–50 min. The mass spectrometer was operated in a data-dependent acquisition mode with a top 10 MS/MS method. MS1 spectra were measured with a resolution of 70,000, an AGC target of $1 \times 10^6$, and a mass range from 350 to 1500 *m/z*. HCD MS/MS spectra were acquired at a resolution of 17,500, an AGC target of $5 \times 10^4$, an isolation window of 2.0 *m/z*, a maximum injection time of 60 ms, and a normalized collision energy of 27. Dynamic exclusion was set to 10 s. Raw data were directly analyzed against the SwissProt database restricted to *Homo sapiens* using Proteome Discoverer version 2.3 (Thermo Fisher Scientific) for identification and label-free precursor ion quantification. The search parameters were as follows: (a) trypsin as an enzyme with up to two missed cleavages; (b) precursor mass tolerance of 10 ppm; (c) fragment mass tolerance of 0.02 Da; (d) carbamidomethylation of cysteine as a fixed modification; and (e) protein N-terminal acetylation, methionine oxidation, and lysine biotinylation as variable modifications. Peptides were filtered at a false-discovery rate of 1% using the percolator node. Normalization was performed such that the total sum of abundance values for each sample over all peptides was the same.

## Statistical analysis

Significant changes were analyzed using a one-way or two-way ANOVA followed by Tukey's post-hoc test using Graph Pad Prism eight software (GraphPad, Inc). For all tests, a *P* value of less than 0.05 was considered statistically significant.

## Acknowledgements

We thank the Applied Protein Research Laboratory of Ehime University. We thank Megumi Kawano for technical assistance. This work was mainly supported by the Platform Project for Supporting

Drug Discovery and Life Science Research (Basis for Supporting Innovative Drug Discovery and Life Science Research [BINDS]) from AMED under Grant Number JP19am0101077 (TS), and by a Grant-in-Aid for Scientific Research on Innovative Areas (JP16H06579 for TS) from the Japan Society for the Promotion of Science (JSPS). This work was also partially supported by JSPS KAKENHI (JP16H04729 and JP19H03218 for TS, and 18KK0229 and 19H04966 to HK) and the Takeda Science Foundation. Additional support for this work was provided by the Joint Usage and Joint Research Programs, the Institute of Advanced Medical Sciences, Tokushima University to TS.

## Additional information

### Funding

| Funder | Grant reference number | Author |
| --- | --- | --- |
| Japan Agency for Medical Research and Development | JP19am0101077 | Tatsuya Sawasaki |
| Japan Society for the Promotion of Science | JP16H06579 | Tatsuya Sawasaki |
| Japan Society for the Promotion of Science | JP16H04729 | Tatsuya Sawasaki |
| Japan Society for the Promotion of Science | JP19H03218 | Tatsuya Sawasaki |
| Japan Society for the Promotion of Science | 18KK0229 | Hidetaka Kosako |
| Japan Society for the Promotion of Science | 19H04966 | Hidetaka Kosako |
| Takeda Science Foundation | | Tatsuya Sawasaki |
| University of Tokushima | Joint Usage Program | Tatsuya Sawasaki |

The funders had no role in study design, data collection and interpretation, or the decision to submit the work for publication.

### Author contributions

Kohki Kido, Data curation, Formal analysis, Validation, Investigation, Methodology, Writing - original draft, Writing - review and editing, Biochemical, molecular, and cellular biology experiments; Satoshi Yamanaka, Validation, Investigation, Assays using CRBN; Shogo Nakano, Data curation, Software, Writing - review and editing, Construction of ancestral BirA proteins; Kou Motani, Investigation, Methodology, Analysis of biotinylated peptides by LC-MS/MS; Souta Shinohara, Investigation, Assay using plant proteins; Akira Nozawa, Methodology, Assay using plant proteins; Hidetaka Kosako, Data curation, Investigation, Methodology, Writing - review and editing, Analysis of biotinylated peptides by LC-MS/MS; Sohei Ito, Resources, Data curation, Software, Writing - review and editing, Construction of ancestral BirA proteins; Tatsuya Sawasaki, Conceptualization, Supervision, Funding acquisition, Methodology, Writing - original draft, Project administration, Writing - review and editing

### Author ORCIDs

Tatsuya Sawasaki https://orcid.org/0000-0002-7952-0556

### Decision letter and Author response
Decision letter https://doi.org/10.7554/eLife.54983.sa1
Author response https://doi.org/10.7554/eLife.54983.sa2

## Additional files

### Supplementary files
• Source code 1. Library-curation toolkit.

• Supplementary file 1. Amino acid and nucleic acid sequences of ancestral BirAs. Amino acid and nucleic acid sequences for the ancestral BirAs designed (AVVA, AFVA, AHLA, GFVA, and all) in this report.

## Data availability

All data generated or analysed during this study are included in the manuscript and supporting files. Source data files have been provided for Figures 3, 4, and 6.

---

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
