## [Decision Letter]

**Acceptance summary:**

This manuscript describes a novel proximity biotinylation enzyme for analysis of protein interactions. This novel enzyme has been completely designed based on ancestral genes. The manuscript describes different in vitro and in vivo assays to show that the new protein is fast but has less toxic effects than established systems such as TurboID. By using different complexes it is demonstrated that the expected interactions are found.

**Decision letter after peer review:**

Thank you for submitting your article "AirID: a novel proximity biotinylation enzyme for analysis of protein-protein interactions" for consideration by *eLife*. Your article has been reviewed by three peer reviewers, including Volker Dötsch as the Reviewing Editor and Reviewer #1, and the evaluation has been overseen by Philip Cole as the Senior Editor. The following individual involved in review of your submission has agreed to reveal their identity: John E. Cronan (Reviewer #3).

The reviewers have discussed the reviews with one another and the Reviewing Editor has drafted this decision to help you prepare a revised submission

Summary:

This manuscript describes a novel proximity biotinylation enzyme for analysis of protein interactions. Biotinylation is still fundamental for molecular level analyses of proteins in vitro and in vivo. The authors successfully constructed a novel enzyme, which has been completely designed based on ancestral genes. The authors use different in vitro and in vivo tests to show that the new protein is fast but has less toxic effects than TurboID. By using different complexes they demonstrate that the expected interactions are found.

Essential revisions:

1) The data of Figure 2C are unexpected and puzzling. The usual scenario is that the mutant BirA makes biotinoyl-5’-AMP that leaks from the active site. If so, it would not be present in the active site to modify the biotin ligation site. Figure 2C shows that addition of the biotin ligation site markedly stimulates AMP production indicating modification of the biotin ligation site. This argues that biotinoyl-5’-AMP remains in the active site rather than being in free solution to give BioID. If another protein (e.g., RelA) replaces the biotin ligation site do we see the same effect?

2) The data presented lack quantitation. Instead only a visual comparisons of a very large number of blots of gels many of which seem overexposed is presented. This paper could be put on a more quantitative basis by an experiment similar to that of the original proximity-dependent biotinylation report (Choi-Rhee, Schulman and Cronan, 2004). Proximity-dependent biotinylation by BirA R118G was detected by biotinylation of many proteins upon its expression in *E. coli*. This simple experiment would allow quantitative assessment of the activities of the various enzymes. Experiments in *E. coli* comparing biotinylation of a fusion protein fused to the enzymes to that seen when the fusion partner in independently expressed would measure biotinylation specificity.

Appropriate controls showing that the enzymes and target proteins are expressed similarly would be required (that is not clear in the present work).

---

## [Author Response]

Essential revisions:1) The data of Figure 2C are unexpected and puzzling. The usual scenario is that the mutant BirA makes biotinoyl-5’-AMP that leaks from the active site. If so, it would not be present in the active site to modify the biotin ligation site. Figure 2C shows that addition of the biotin ligation site markedly stimulates AMP production indicating modification of the biotin ligation site. This argues that biotinoyl-5’-AMP remains in the active site rather than being in free solution to give BioID. If another protein (e.g., RelA) replaces the biotin ligation site do we see the same effect?

We thank the reviewer for bringing up excellent comments. From the result of Figure 2C, we considered that GFVA-RG and GFVA-RS (AirID) have two abilities: biotinylation on bls with its site recognition, and release of free biotinoyl-5’-AMP. However, in this figure, we would indicate whether AirID has the ability for releasing biotinoyl-5’-AMP, but not whether AirID can biotinylate on bls (Avitag). Pointing out from this reviewers’ comment, we thought that this figure provides the extra confusion. Therefore, we changed Figure 2C data into Figure 2—figure supplement 4C showing that AirID has the ability for releasing biotinoyl-5’-AMP.

2) The data presented lack quantitation. Instead only a visual comparisons of a very large number of blots of gels many of which seem overexposed is presented. This paper could be put on a more quantitative basis by an experiment similar to that of the original proximity-dependent biotinylation report (Choi-Rhee, Schulman and Cronan, 2004). Proximity-dependent biotinylation by BirA R118G was detected by biotinylation of many proteins upon its expression in *E. coli*. This simple experiment would allow quantitative assessment of the activities of the various enzymes. Experiments in *E. coli* comparing biotinylation of a fusion protein fused to the enzymes to that seen when the fusion partner in independently expressed would measure biotinylation specificity.Appropriate controls showing that the enzymes and target proteins are expressed similarly would be required (that is not clear in the present work).

We thank the reviewer for bringing up this constructive comment. We think this is an important issue for our method. Some eukaryotic full-length proteins used in our reports, such as RelA and SALL4, cannot be synthesized by *E. coli*. Therefore, we cannot carry out their experiments using *E. coli* cells. Instead of them, quantified band intensities were showed in each main figure, Figure 1C, D, Figure 2A, B, D, Figure 3D, E, and Figure 4C.

About expression of the enzymes and target proteins, in all figures, both enzymes and target proteins were detected using antibody recognized fusing tag or protein itself. In each figure legend below, we added sentences concerning each appreciate control.

Figure 1C and D

Figure 2A, B and D

Figure 2—figure supplement 1

Figure 2—figure supplement 2

Figure 2—figure supplement 3

Figure 2—figure supplement 4B

Figure 2—figure supplement 5

Figure 2A, D and E

Figure 3—figure supplement 1

Figure 3—figure supplement 2

Figure 4A, B and C

Figure 4—figure supplement 1